# Erythroid Differentiation Regulator 1 Ameliorates Collagen-Induced Arthritis via Activation of Regulatory T Cells

**DOI:** 10.3390/ijms21249555

**Published:** 2020-12-15

**Authors:** Myun Soo Kim, Sora Lee, Sunyoung Park, Kyung Eun Kim, Hyun Jeong Park, Daeho Cho

**Affiliations:** 1Korea University Kine Sciences Research Institute, Kine Sciences, 525, Seolleung-ro, Gangnam-gu, Seoul 06149, Korea; mskim@kinesciences.com (M.S.K.); srlee@kinesciences.com (S.L.); sypark@kinesciences.com (S.P.); 2Department of Cosmetic Sciences, Sookmyung Women’s University, Cheongpa-ro 47-gil 100 (Cheongpa-dong 2ga), Yongsan-gu, Seoul 04310, Korea; kyungeun@sookmyung.ac.kr; 3Institute of Convergence Science, Korea University, Anam-ro 145, Seongbuk-gu, Seoul 02841, Korea; dermacmc@naver.com

**Keywords:** Erythroid differentiation regulator 1, regulatory T cells, rheumatoid arthritis

## Abstract

Erythroid differentiation regulator 1 (Erdr1) has been identified as an anti-inflammatory factor in several disease models, including collagen-induced arthritis (CIA), but its exact mechanisms are still not fully understood. Here, the involvement of regulatory T (Treg) cells in Erdr1-improved CIA was investigated. In the CIA model, Erdr1 was confirmed to reduce collagen-specific IgM in plasma and plasma cells in draining lymph nodes. Importantly, the downregulated Treg cell ratio in draining lymph nodes from CIA mice was recovered by Erdr1 treatment. In addition, administration of Erdr1 improved the CIA score and joint tissue damage, while it revealed no effect in Treg cell-depleted CIA mice, indicating that Treg cells mediate the therapeutic effects of Erdr1 in the CIA model. Results from in vitro experiments also demonstrated that Erdr1 significantly induced Treg cell differentiation and the expression of Treg activation markers, including CD25, CD69, and CTLA4 in CD4^+^Foxp3^+^ cells. Furthermore, Erdr1-activated Treg cells dramatically suppressed the proliferation of responder T cells, suggesting that they are functionally active. Taken together, these results show that Erdr1 induces activation of Treg cells and ameliorates rheumatoid arthritis via Treg cells.

## 1. Introduction

Regulatory T (Treg) cells are major immune-suppressive lymphocytes that have a pivotal role in immune homeostasis by maintaining immune tolerance, and they are commonly characterized as CD4^+^ T cells highly expressing CD25 and Fork head box P3 (Foxp3). Because of these regulatory abilities, there are various efforts to induce functional Treg cells as a therapeutic method for inflammatory diseases such as autoimmune diseases and allergy [1,2]. In addition, crucial molecules for functionally suppressive Treg cells, including CD25, CD69, and CTLA4 as well as Foxp3, have been identified [3,4,5]. High expression of CD25 on Treg cells results in rapid consumption of IL-2, thereby restraining activities of the other lymphocytes [6]. Treg cells also downregulate costimulatory molecules, such as CD80 and CD86, on dendritic cells via highly expressed CTLA4 [7]. Furthermore, it has been reported that CD69 expression on Treg cells is essential for maintenance of Treg cell functions [3].

Erythroid differentiation regulator 1 (Erdr1) has been identified as a stromal survival factor released in stressful conditions [8], and various anti-inflammatory activities of Erdr1 were also reported. The negative correlation between Erdr1 and IL-18 expression has been shown [9], and therapeutic effects of Erdr1 on inflammatory diseases such as rheumatoid arthritis (RA), psoriasis, and rosacea have been determined [10,11,12]. Moreover, it has recently been shown that Erdr1 controls T cell receptor (TCR)-mediated Ca^2+^ signaling in thymocytes [13]. However, the therapeutic mechanisms of inflammatory diseases and the roles of Erdr1 in T cell immunology are not well understood.

In the current study, the therapeutic mechanisms of Erdr1 in collagen-induced arthritis (CIA) were investigated. Erdr1 treatment was confirmed to improve CIA as reported previously, and Erdr1 also recovered Treg cell population, which was decreased in draining lymph nodes after arthritis induction. Moreover, in the Treg-depleted condition, Erdr1 failed to reduce arthritis, suggesting that the effects of Erdr1 on CIA were mediated by Treg cells. In vitro results also showed that Erdr1 significantly induced the expression of Treg activation markers, CD25, CD69, and CTLA4 in CD4^+^Foxp3^+^ Treg cells in the presence of TCR stimuli. In addition, the Erdr1-activated Treg cells downregulated T cell proliferation, indicating that the Erdr1-treated cells are functionally suppressive.

## 2. Results

### 2.1. Erdr1 Ameliorates the Experimental Rheumatoid Arthritis via Treg Cells

The anti-inflammatory activity and therapeutic effects of Erdr1 on the CIA model have previously been reported [10]. In this study, Erdr1 was confirmed to reduce the plasma levels of bovine type II collagen (CII)-specific IgM, and Erdr1 also downregulated CD138^+^ plasma cells in draining lymph nodes (Figure 1A,B). Interestingly, Treg cells, which were decreased in CIA mice, were recovered by Erdr1 treatment (Figure 1C), suggesting that Treg cells could mediate the effects of Erdr1 on the improvement of CIA. To examine this hypothesis, a Treg cell depletion assay was performed in the CIA model using anti-CD25 antibody [14]. Erdr1 was treated one day post-depletion, and more than 50% of Treg cells in peripheral lymph nodes were depleted after 48 h from the antibody injection (Appendix A). Surprisingly, Erdr1 failed to improve arthritis in Treg-depleted mice, while the disease score was significantly decreased in control antibody-treated mice (Figure 1D,E). In addition, tissue section analyses, including H&E and safranin O stains, revealed that Erdr1 reduced damages of articular cartilage in the CIA model, whereas it had no effect on Treg-depleted CIA mice (Figure 2). These results indicate that Treg cells are involved in the amelioration of CIA by Erdr1.

### 2.2. Erdr1 Induces Activation of Treg Cells

To examine the effects of Erdr1 on Treg cell activation, primary CD4 T cells were isolated from peripheral lymph nodes of mice and cultivated with or without Erdr1 in the absence or presence of TCR stimulation. Interestingly, CD69^+^Foxp3^+^ Treg cells were induced significantly by Erdr1 in the presence of TCR stimuli (Figure 3A). It is worth noting that CD69 is regarded as an immunoregulatory molecule, and the expression of CD69 on Treg cells is required for maintenance of suppressive Treg functions [3,15]. Furthermore, expressions of CD25 and CTLA4, two other functional Treg markers [4,5], were also significantly increased on Treg cells by Erdr1 (Figure 3B), while cell death was not increased in this condition of T cells as well as Treg cells (Appendix A). These results demonstrate that Erdr1 induces activation of Treg cells.

### 2.3. Erdr1-Activated Treg Cells are Functionally Suppressive

Although Erdr1 induced the expression of activation markers on Treg cells (Figure 3), Foxp3^+^ non-suppressive T cells have been reported [16], and the regulatory functions of Treg cells can be lost under various conditions [17]. Therefore, it was necessary to confirm whether the Erdr1-treated Treg cells were functionally active. To determine suppressive functions of Erdr1-activated Treg cells, CFSE-labeled responder T cells were co-cultivated with media- or Erdr1-treated T cells, and the proliferation of responder T cells was evaluated in the presence of TCR stimulation. Results revealed that the Erdr1-treated T cells suppressed the proliferation of responder T cells dramatically, and the ratio of the undivided cells (M1 marker) was significantly higher in responders incubated with Erdr1-treated T cells (Figure 4A,B). These data show that Erdr1-activated Treg cells are functionally suppressive.

## 3. Discussion

In the present study, rheumatoid arthritis was improved by Erdr1, while the therapeutic effects were lost in Treg-depleted CIA mice, suggesting that Treg cells are required for the Erdr1-induced amelioration of the disease. Furthermore, Erdr1 induced the activation of Treg cells with increased expression of CD25, CD69, and CTLA4. Especially, the Erdr1-activated Treg cells efficiently inhibited the proliferation of responder T cells, indicating that the Treg cells were functionally suppressive.

Although Erdr1 is hard to detect in serum, probably due to the low sensitivity of Erdr1 antibodies, it is worthwhile to examine the blood levels of Erdr1 in the CIA mice. Erdr1 previously showed anti-inflammatory functions and a negative correlation with IL-18 expression in synovial and skin tissues [9,10,11,12]. As the expression of Il-18 is elevated in RA tissues [18,19], blood concentrations of Erdr1 are thought to be decreased in the CIA mice. To prove this hypothesis, detection methods for serum Erdr1 should be improved in further studies.

Various cytokines, including IL-2, IL-10, and TGFb, have been reported to induce Treg function and activation [1,20]. However, the molecular mechanism of Erdr1 for Treg activation is thought to be unlike those cytokines. It is well-established that IL-2 promotes T cell survival [21], whereas Erdr1 showed dual functions of T cell viability (Appendix A) [22]. IL-10 does not affect expression of CD25 and CD69 [23]. TGFb also has differential effects on levels CD25 and CD69 [24], while Erdr1 enhances the expression of both molecules, indicating that Erdr1 activates Treg cells via a different signal pathway from these cytokines. In a recent study, IFNa has been shown to induce Treg differentiation [25]. The effects of IFNa on Treg cells are similar to Erdr1 because IFNa induces surface expression of CD25 and CD69 on Treg cells [26]. In addition, IFNa has reduced antigen-induced arthritis by Treg mediation [27]. However, Erdr1 and IFNa revealed opposite functions in the production of inflammatory cytokines. IFNa is known to promote inflammatory cytokines, such as IL-18, against microbial infections [28], while Erdr1 has shown negative correlation to the expression of IL-18 and anti-inflammatory activities [9,10,11,12]. Therefore, further cellular and molecular mechanisms, including identification of receptors, should be uncovered in order to better understand how Erdr1 affects Treg activation.

Erdr1 has been reported to be expressed in various tissues, including immune organs such as the spleen, bone marrow, and thymus, and to affect thymocytes selection via regulation of TCR-mediated Ca^2+^ influx [13]. In the current study, Erdr1 also enhanced Treg activation from isolated CD4 T cells with TCR stimulation (Figure 3), suggesting that Erdr1 can regulate T cell responses directly, as well as T cell development. Although Erdr1 was shown to promote apoptosis of T cells as an autocrine [22], the ratio of live cells was increased by Erdr1 in the presence of TCR stimulation (Appendix A). In addition, Erdr1-induced cell death was observed at concentrations of 10 and 100 ng/mL, while cell viability was increased with 1000 ng/mL of Erdr1 in the absence of TCR stimulus (Appendix A), indicating that Erdr1 engages in T cell immunology differently depending on Erdr1 doses and presence of TCR signal. These results also suggest that Erdr1 is a critical regulator for T cell immunology, which controls thymocytes selection [13], T cell death [22], and Treg activation.

The involvement of Erdr1 in Treg activation, such as enhancement of CD25, CD69, and CTLA4 expression, is especially important, as this ability can be applied to various inflammatory diseases as well as RA. Transfer of CD69^+^ Treg cells to asthmatic mice has restored immune tolerance [3], and injection of CD25^hi^ Treg cells has also improved hepatic ischemia-reperfusion injury [4]. In addition, CTLA4^+^ Treg cells have been reported to prevent inflammatory tissue attack in CIA mice [29]. It is worth noting that Erdr1 enhances Treg activation by induction of CD25 and CTLA4 expression, since the expressions of CD25 and CTLA4 are decreased in Treg cells from RA patients [30,31]. Therefore, application of Erdr1 for inflammatory diseases should be considered as a modulator for T cell immunity.

## 4. Materials and Methods

### 4.1. Mice and Cells

Six-week-old male DBA/1J (Central Lab. Animal Inc., Seoul, Korea) mice were used for the CIA model, and 7- to 10-week-old male DBA/1J mice were used for in vitro experiments using primary CD4 T cells. The animals were housed in a specific pathogen-free facility, and all experiments were approved by the Korea University Institutional Animal Care and Use Committee (KUIACUC-2018-0024 approval date: 12 October 2018, 2019-0088 approval date: 16 September 2019, renewal date: 18 August 2020). Murine primary CD4 T cells were isolated from lymph nodes of mice and cultivated at 37 °C in 5% CO_2_ using RPMI 1640 media (WelGENE Inc, Daegu, Korea) with FBS (10%, WelGENE Inc), penicillin (100 U/mL, Invitrogen, Carlsbad, CA, USA), streptomycin (0.1 mg/mL, Invitrogen, Carlsbad, CA, USA), and 2-ME (50 μM, Sigma-Aldrich, St Louis, MO, USA).

### 4.2. Collagen-Induced Arthritis Model and Treg Depletion

The collagen-induced arthritis model was set up as previously described, with eight mice per group [10]. Equal volumes of bovine type II collagen (CII) and complete Freund’s adjuvant (Chondrex, Redmond, WA, USA) were mixed, and male DBA/1J mice were immunized with the emulsion (CII, 40 μg/mouse) via intradermal injection into the tail. Fourteen days post immunization, Treg-depletion antibody (PC-61.5.3) or isotype-control antibody were i.p. injected once into mice (250 mg/mouse, BioXCell, Western Lebanon, NH, USA), and mice were boosted with an emulsion of CII (40 μg/mouse) in incomplete Freund’s adjuvant (Chondrex, Redmond, WA, USA) with 6 h interval from the depletion-antibody injection. One day after boosting, PBS, Erdr1 (0.1 mg/kg), or methotrexate (MTX, 1 mg/kg) was injected (i.p., 3 times/week) for 3 weeks. Arthritis score and CII-specific IgM from plasma were evaluated as previously described [10].

### 4.3. Antibodies and Recombinant Erdr1

Alexa488-conjugated anti-Foxp3 (MF23), PerCP-Cy5.5-conjugated anti-CD4 (RM4-5), FITC-conjugated anti-CD25 (3C7), PE-conjugated anti-CD138 (281-2), and anti-CD3ε (145-2C11) were purchased from BD Biosciences (San Diego, CA, USA). PE-conjugated anti-CTLA4 (UC10-4B9), APC-conjugated anti-Foxp3 (FJK-16s), and APC-conjugated anti-CD69 (H1.2F3) were obtained from eBioscience (San Diego, CA, USA).

Recombinant Erdr1 was prepared as described previously [32]. Briefly, the bacterial vector with the CDS region of Erdr1 was prepared from the Erdr1-pCMV-SPORT6 plasmid (Open Biosystems, Huntsville, AL, USA). Erdr1 with more than 95% purity was purified from bacteria. The endotoxin level was examined by the *Limulus* amebocyte lysate assay (Cape Cod, East Falmouth, MA, USA). Low endotoxin (<0.1 EU/mL)-contained lots were used for experiments.

### 4.4. Tissue Stain

Mice feet were fixed in 4% paraformaldehyde and embedded in paraffin. Hematoxylin and eosin (H&E) staining with tissue sections was performed using standard techniques [33]. For safranin O staining, deparaffinized slides were rehydrated and incubated with Hematoxylin QS solution (Vector Laboratories, Burlingame, CA, USA) for 5 min. After washing in running tap water for 5 min, slides were destained quickly in acid ethanol and washed again. Staining with 0.001% Fast green FCF solution (Sigma-Aldrich, St Louis, MO, USA) was conducted for 5 min, and slides were rinsed with 1% acetic acid solution for 10 s. After staining with 0.2% safranin O solution (Merck Millipore, Burlington, MA, USA) for 5 min, slides were dehydrated and mounted. All experimental processes were performed at room temperature.

### 4.5. T Cell Stimulation and Suppression of T Cell Proliferation

Murine CD4 T cells were isolated from lymph nodes of mice using CD4 T cell isolation kit and MACS system (Miltenyi Biotec, Auburn, CA, USA). CD4 T cells (1 × 10^5^ cells/well) were cultured with various concentrations of Erdr1 in an anti-CD3ε antibody (0, 0.25, or 0.5 mg/mL)-coated 96-well plate for 48 h in triplicate.

For the T cell proliferation assay, media- or Erdr1-treated (1 mg/mL) CD4 T cells in the presence of TCR stimulation (48 h cultivation) were used as effector cells. CFSE-labeled (1 mM) responder CD4 T cells (1 × 10^5^ cells/well) were co-cultured with the effector cells (1:1 ratio) in an anti-CD3ε antibody (0.5 mg/mL)-coated 96-well plate for 3 days in duplicate.

### 4.6. Flow Cytometry

Single-cell suspensions were prepared and stained with antibodies at 1:250 to 1:500 dilution in FACS buffer. Cell surface staining was performed for 15 min. Fixation and intracellular stains were done using Cytofix/Cytoperm and Perm/Wash solutions (BD Biosciences, San Diego, CA, USA). Non-specific binding was monitored using fluorescent-conjugated control antibodies. Every flow cytometric analysis was performed with live cell gates, using FACSCalibur and CellQuest or FlowJo software (BD Biosciences, San Diego, CA, USA).

### 4.7. Statistical Analyses

A non-paired and two-tailed Student’s *t*-test was used to compare control and experimental groups. *p*-values < 0.05 among results with Cohen *d* > 3 were considered to be statistically significant.

## Figures and Tables

**Figure 1 ijms-21-09555-f001:**
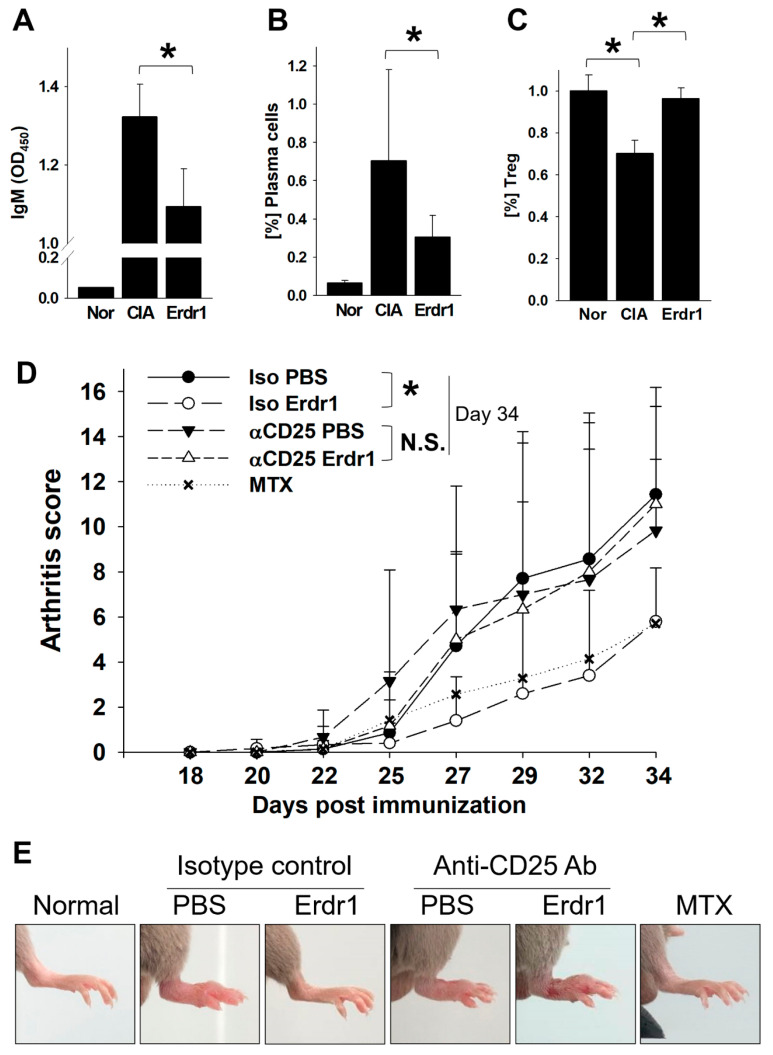
Regulatory T (Treg) cells mediate the therapeutic effects of Erythroid differentiation regulator 1 (Erdr1) on the collagen-induced arthritis (CIA) model. The CIA model was established as described in Materials and Methods. (**A**) Levels of CII-specific IgM were examined from plasma after 1 week of Erdr1 treatment using ELISA. The population of CD138^+^ plasma cells (**B**) or Treg cells (**C**) in draining lymph nodes from mice treated with Erdr1 for 3 weeks was evaluated by flow cytometric analysis. (**D**,**E**) Fourteen days after the first immunization of the CIA model, Treg-depletion antibody or isotype-control antibody was injected (250 mg/mouse), and mice were boosted with 6 h interval. One day after boosting, PBS, Erdr1 (0.1 mg/kg), or methotrexate (MTX, 1 mg/kg) was administered (i.p., 3 times/week) for 3 weeks. Arthritis score (**D**) and joint swelling (**E**) were compared from the Treg-depleted CIA mice. Eight mice per group were used for the RA model study. * *p* < 0.05.

**Figure 2 ijms-21-09555-f002:**
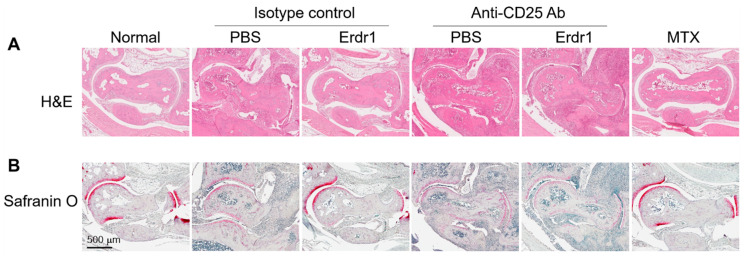
Treg cells involved in the improvement of cartilage damage by Erdr1. Hind-limb tissues from the Treg-depleted CIA experiment were fixed in 4% paraformaldehyde, and paraffin section was performed. After deparaffinization, (**A**) the tissue samples were stained with H&E, and (**B**) cartilages were colored (red) with safranin O (0.2%) at RT. Scale bar = 500 µm.

**Figure 3 ijms-21-09555-f003:**
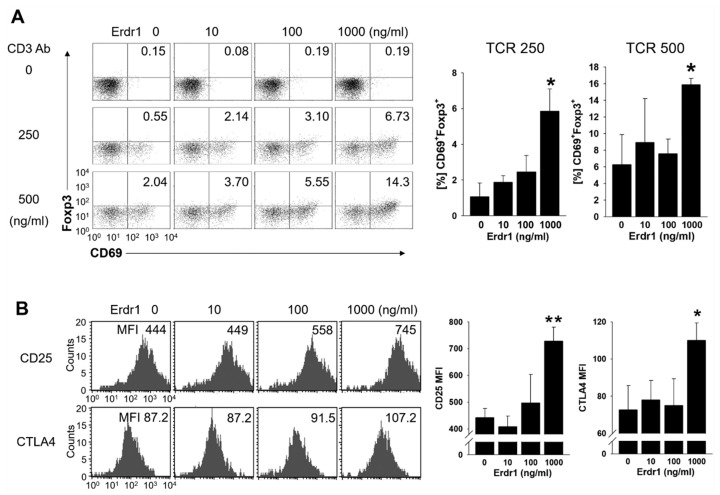
Erdr1 enhances the expression of activation markers in Treg cells. CD4 T cells were isolated from peripheral lymph nodes of DBA1 mice and cultivated with or without Erdr1 in the presence of T cell receptor (TCR) stimuli for 48 h. (**A**) CD4 T cells were cultured in anti-CD3ε antibody-coated plates (0, 250, 500 ng/mL), and Foxp3^+^CD69^+^ cells were evaluated by flow cytometry with CD4^+^ gating. Results were summarized as mean ± SD from three independent experiments in triplicate (*n* = 3). (**B**) CD4 T cells were treated with Erdr1 in the presence of anti-CD3ε antibody (250 ng/mL) for 48 h. The expression levels of the Treg activation markers CD25 and CTLA4 were examined using flow cytometric analysis, and Mean fluorescence intensity (MFI) data were summarized as mean ± SD from three independent experiments in triplicate (*n* = 3). Cells were gated on CD4^+^FoxP3^+^. The numbers indicate the percentages of quadrants in (**A**) and MFI values in (**B**), respectively. The flow cytometric results are representative of data from three independent experiments. * *p* < 0.05, ** *p* < 0.01 (vs. Erdr1 0 ng/mL).

**Figure 4 ijms-21-09555-f004:**
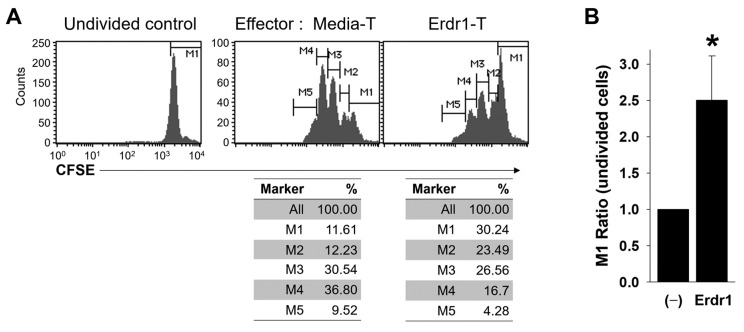
Erdr1-activated Treg cells are functionally suppressive. CD4 T cells from lymph nodes of DBA1 were cultured with media alone (Media-T) or Erdr1 (1 μg/mL, Erdr1-Treg) in the presence of TCR stimuli and used as effector cells. CFSE-labeled (2 mM) responder CD4 T cells were co-cultivated with the effector cells (1:1 ratio) in the presence of anti-CD3ε antibody (0.5 μg/mL) for 3 days. (**A**) Proliferation of responder cells (gated on CFSE^+^ cells) were determined by flow cytometry. (**B**) Ratios of the undivided cells (M1) were compared and presented as mean ± SD from three independent experiments in duplicate (*n* = 3). * *p* < 0.05 (vs. Media-T control).

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
