# Peer review of "Erythroid Differentiation Regulator 1 Ameliorates Collagen-Induced Arthritis via Activation of Regulatory T Cells"

_ijms, 2020, doi:10.3390/ijms21249555_

Round 1

Reviewer 1 Report

Scientific paper of high interest, clearly written, with interesting rationale and well-conducted and described experiments. The hypotheses are clearly supported by the results of experiments. Acceptable without revisions.

Author Response

Reviewer 1 had no comment.

Reviewer 2 Report

This is an interesting study about the role of  Erdr1 in activation of Treg cells and his contribution to ameliorates rheumatoid arthritis via  Treg cells, and entitle “Erythroid Differentiation Regulator 1 Ameliorates 2 Collagen Induced Arthritis via Activation of Regulatory  T Cells”

The paper is well written and the experiments are well conducted.

However I’ve some clarification to ask the authors.

1) In the figure 1E the authors showed the effect of MTX on joint swelling without discuss this results. I’ve two curiosity: which is the serum concentration of Erdr1 in CIA model and  what happens to Erdr1 concentration after MTX treatment?

2) In the discussion section the authors speculate regarding the pathway involved in Treg activation by Erdr1, specifying that it is certainly different from the pathway activated by cytokines. But, in your opinion,  which could it be?

3) Always in the discussion section, the authors talk about the different effects of Rdr1 based on its concentration. what happens to Treg apoptosis using your experimental concentration?

4) Finally, in the title the authors emphasize the role of Erdr1 in ameliorates of rheumatoid arthritis. However, in the discussion this important point is poor. Can the authors to expand this concept?

Minor:

Symbols mistakes in line 210,212,215

Author Response

  1. (Comment): In the figure 1E the authors showed the effect of MTX on joint swelling without discuss this results. I’ve two curiosity: which is the serum concentration of Erdr1 in CIA model and what happens to Erdr1 concentration after MTX treatment?

(Response): The reviewer’s comment is a good point and we are also curious about the blood level of Erdr1 from those mice. Unfortunately, ELISA for serum Erdr1 is not available so far and blotting methods are not sensitive to detect serum Erdr1. Since the expression of Erdr1 showed negative correlation with IL-18 expression (Jung, et al., 2011, J Invest Dermatol), which is elevated in RA tissues (Dayer, 1999, J Clin Invest; Gracie, et al., 1999, J Clin Invest), serum concentration of Erdr1 is thought to be decreased in CIA mice. We added this point of view in the “Discussion” section (line 136-141).

  1. (Comment): In the discussion section the authors speculate regarding the pathway involved in Treg activation by Erdr1, specifying that it is certainly different from the pathway activated by cytokines. But, in your opinion, which could it be?

(Response): This comment is also an important point. As we mentioned in the “Discussion”, biological phenomena of Erdr1-treated T cells are different from those of IL-2, IL-10, or TGFb treated cells. Instead, Erdr1 was previously shown to regulate TCR-mediated Ca2+ influx in thymocytes (Kim, et al., 2019, Cell Immunol), which can be controlled by TCR signaling complex. Because the Ca2+ flux changes were not observed without TCR stimulus, Erdr1signaling seems to be different from typical cytokines, but rather related with TCR modulators at least in T cells. Further studies will be proceeded to prove this hypothesis.

  1. (Comment): Always in the discussion section, the authors talk about the different effects of Rdr1 based on its concentration. what happens to Treg apoptosis using your experimental concentration?

(Response): As the reviewer commented, Treg cell survival was evaluated from cells gated on CD4+Foxp3+ in the absence or presence of TCR stimulation. The Treg survival results were added in the Supplementary Figure 2 and the “Results” section (Figure S2C; line 97).

  1. (Comment): Finally, in the title the authors emphasize the role of Erdr1 in ameliorates of rheumatoid arthritis. However, in the discussion this important point is poor. Can the authors to expand this concept?

(Response): As the reviewer commented, we improved the discussion about the role of Erdr1 as a potent Treg activator in inflammatory diseases (line 168-175).

Minor:

Symbols mistakes in line 210,212,215

(Response): We appreciate the reviewer’s comment, the symbols were corrected (line 222-227).

Reviewer 3 Report

Very interesting study, with solid methodology.

A few concerns :

The sample sizes should be indicated in ALL the figure fonts and further discussed in the Methods section. 

Figure 3 : is the sample size sufficient? Please justify.

Were all the experiments performed in duplicate? triplicate?

Author Response

Reviewer #3

  1. (Comment): The sample sizes should be indicated in ALL the figure fonts and further discussed in the Methods section.

(Response): As the reviewer commented, sample sizes, such as animal numbers for the RA model and repeated experimental numbers, were added in the “Materials and Methods” section and each figure legend (legends of Figure 1,3, and 4; line 188-189).

  1. (Comment): Figure 3: is the sample size sufficient? Please justify.

(Response): When we calculated effect size values (Cohen’s d) of data from Figure 3, all values are high with comparison between the “Erdr1 0 ng/ml” group and the “Erdr1 1000 ng/ml” group (Table below), as well as low p values (p < 0.05). Therefore, we concluded the sample size (three independent experiments) is sufficient at least to compare those two groups (Erdr1 0 vs 1000 ng/ml). We added the effect size values in the “Materials and Methods” section and figure legends (line 237).

Figure

Data

Cohen’s d

A. TCR250

[Erdr1 0 ng/ml] vs [Erdr1 1000 ng/ml]

4.668

A. TCR500

[Erdr1 0 ng/ml] vs [Erdr1 1000 ng/ml]

3.674

B. CD25

[Erdr1 0 ng/ml] vs [Erdr1 1000 ng/ml]

6.571

B. CTLA4

[Erdr1 0 ng/ml] vs [Erdr1 1000 ng/ml]

3.288

  1. (Comment): Were all the experiments performed in duplicate? triplicate?

(Response): All in vitro experiments were performed in duplicate (for Figure 4) or triplicate (for Figure 3) and repeated total three times (three independent experiments were performed). We mentioned these in each figure legend and the “Materials and Methods” section (legends of Figure 3 and 4; line 223; line 227).

Round 2

Reviewer 3 Report

My revision requests were all fulfilled.

The article is now fit for publication in my opinion.